# Buffering Copper Tailings Acid Mine Drainage: Modeling and Testing at Fushë Arrëz Flotation Plant, Albania

**Giuseppe Cocomazzi** [1,*], **Giovanni Grieco** [1,*], **Agim Sinojmeri** [2], **Alessandro Cavallo** [3], **Micol Bussolesi** [3], **Elena Silvia Ferrari** [1] and **Enrico Destefanis** [4]

1   Department of Earth Sciences, University of Milan, 20133 Milan, Italy; elena.ferrari@unimi.it
2   Department of Earth Sciences, Faculty of Geology and Mining, Polytechnic University of Tirana, 1010 Tirana, Albania; agim.sinojmeri@fgjm.edu.al
3   Department of Earth and Environmental Sciences—DISAT, University of Milan-Bicocca, 20126 Milan, Italy; alessandro.cavallo@unimib.it (A.C.); micol.bussolesi@unimib.it (M.B.)
4   Department of Earth Sciences, University of Turin, 10125 Turin, Italy; enrico.destefanis@unito.it
*   Correspondence: giuseppe.cocomazzi@unimi.it (G.C.); giovanni.grieco@unimi.it (G.G.)

**Abstract:** The beneficiation process of sulfide ores has the inevitable consequence of generating huge amounts of tailings highly enriched in sulfur, thus inducing acid mine drainage (AMD) and the release of potentially toxic elements. The aim of the work was to define the most suitable procedures for buffering acid drainage waters through the addition of commercial $CaCO_3$ paste, provided by UNICALCE. High- and low-pyrite tailing samples were collected at the copper enrichment plant of Fushë Arrëz (Northern Albania copper mining district). They were used for leaching and buffering tests, whose leachates and precipitation products were characterized by ICP-MS, chromatographic, XRD and TEM analyses. In addition, a geochemical model was developed in order to predict the pH trend of the leachate as a function of the addition of $CaCO_3$. The results show the good buffering capacity of $CaCO_3$, accurately predicted by the geochemical model. A drastic reduction in metals in the solution can be easily attained for low-pyrite samples, whereas high amounts of buffering agent are required to reach similar metals concentration reduction in high-pyrite. Precipitates are dominated by oxyhidroxides, followed by sulfates and hydrosilicates, but TEM showed also the presence of nanocrystalline and amorphous phases.

**Keywords:** geochemical modeling; acid drainage; buffering; metal precipitation

## 1. Introduction

For more than 40 years, Albania has been the site of intense mining activity linked to numerous deposits and enrichment plants for chromium, copper, coal and iron–nickel concentrates.

As a result of the extraction process from metal deposits, active or abandoned mining plants have accumulated enormous amounts of solid waste over the decades. These materials can be very rich in sulfides and, if not properly treated, constitute a considerable environmental risk. The Acid Mine Drainage (AMD) and the Potentially Toxic Elements (PTE) release are two of major sources of pollution in sulfide-rich mining sites and are an expensive burden to manage. AMD is a process produced when sulfide minerals are exposed to oxidizing conditions due to mining activities. Among the sulfides, iron ones (particularly pyrite) are the main producers of AMD. When pyrite is exposed to water and oxygen, it transforms into dissolved iron, sulfate and hydrogen, increasing the acidity of the water [1–3]. The acid drainage from mines depends on the acid-producing (sulfide) and buffering (mainly from natural such as carbonates to constructed such as permeable barriers of various materials [4,5]) minerals exposed to the atmosphere.

In general, sulfide-rich and carbonate-poor materials are expected to produce acid drainage, whereas the presence of abundant carbonate materials, even with significant amounts of sulfides, is more likely to produce waters with a higher pH [6,7].

Predicting the evolution of acid mine drainage (AMD) is of increasing interest to the mining industry due to its potential for long-term environmental damage. The tests typically used to predict the evolution of AMD are both laboratory- and model-based, with standardized methods of static and kinetic laboratory testing often being applied [8,9].

Static (acid-base accounting) tests are designed to compare the acid-generating potential and acid-neutralizing potential of mining wastes and tailings, but in order to predict the long-term acid generating potential, kinetic tests are more reliable [10], but are associated with a high degree of uncertainty. Furthermore, these methods largely neglect the crucial interaction with oxygen availability [11]. Modeling studies of AMD generation focus on the examination of the effects of dominant processes and compare different tailing remediation measures [12–14]. Nevertheless, the predictive capability of many of these models is limited [8,15], due to, for example, over simplification or neglect of some important geochemical processes.

In addition to several other factors affecting AMD, the specific enrichment procedure can play a major role in the potential acidity of dumped tailings. Previous studies show that a different combination of flotation processes results in tailings with quite different environmental impact [16,17] and with high but still variable acidification capacity as a response to leaching tests [16].

This work aims to define the most suitable procedures for the buffering of leached waters from AMD producing tailing materials. The precipitation of heavy metals as a result of the buffering was also investigated and major, minor and trace elements distribution was determined (e.g., [7,18–23] and references therein).

The optimization of the buffering procedure is based on the acidity and metal contents of the buffered waters that must meet the surficial waters quality requirements and the minimization of the amount of sludge formed due to the precipitation of ions in solution during the pH variation, sludge that requires a final disposal as a special waste at high costs.

The Fushë Arrëz enrichment plant works copper ore coming from the Munella and Lak Roshi mines, within the Volcanic Massive Sulfide (VMS) copper mining district of northern Albania (Figure 1), where up to 25 mines and three enrichment plants were active in the past at the same time. This district is located within the Eastern Ophiolite Belt of the Mirdita ophiolite and is mainly enriched in chalcopyrite and pyrite, with smaller amounts of bornite, sphalerite, covellite, chalcocite and occasionally tennantite and arsenopyrite (e.g., [18,24,25]). The main sulfide related metals are Cu, Zn, Fe, Au, associated with variable contents of Pb, Ag, As, Cd, Ge, Sb, In [26–28].

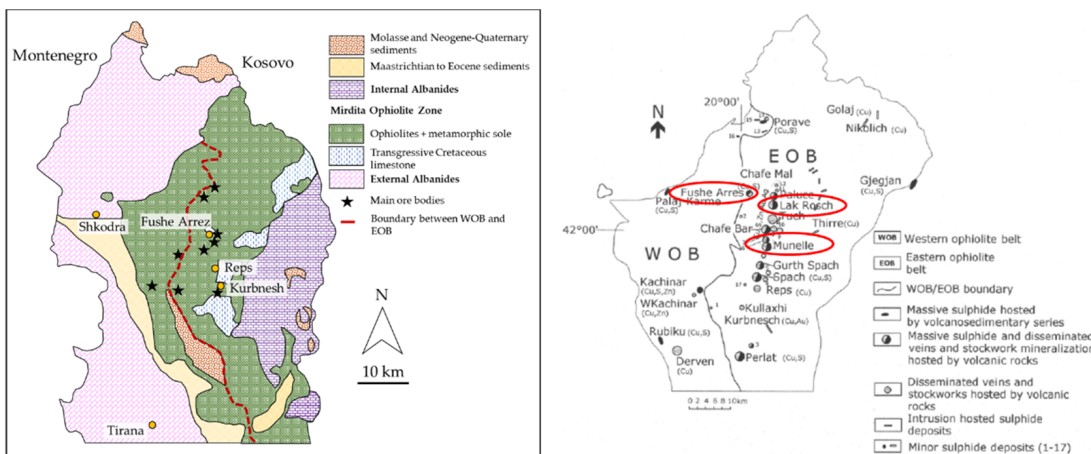

**Figure 1. On the left**: simplified geological map of the Albanian ophiolites and surrounding units (modified from [29]). The dashed line depicts the approximate boundary between the western- and eastern-type ophiolites. **On the right**: spatial distribution of major sulfide mineralized bodies in north-central Albanian ophiolites [26], whereas the Fushë Arrëz enrichment plant and the two copper ore mines Munella and Lak Roshi are marked in red circles.

## 2. Materials and Methods

We here investigated the same tailing dumps collected and partially studied by Grieco et al. [16] at the Fushë Arrëz dressing plant. As reported by Grieco et al. [16], samples were assembled to be the most representative possible of each dump material. The two dumps cover an area of approximately $6 \times 10^4$ m$^2$ (FA4) and $18 \times 10^4$ m$^2$ (FA3), respectively. The extreme morphological variability of the earth surface and the altitudes shown on maps (Google Earth), which vary by approximately 30–40 m from mountain to valley, allow a very rough estimate of the dump volumes. It can be assumed that they are approximately $12 \times 10^5$ m$^3$ (FA4) and $27 \times 10^5$ m$^3$ (FA3), respectively. A semi-quantitative determination of mineralogy from XRD data is reported in Table 1:

**Table 1.** Semi-quantitative determination of mineralogical composition of new dump and old dump (data in wt%), Grieco et al. [16].

| Sample | Pyrite | Quartz | Gypsum | Chlorite |
|---|---|---|---|---|
| FA3 New dump | 68 | 23 | 3 | 6 |
| FA4 Old dump | 18 | 34 | | 48 |

At this enrichment plant, 2000 pyrite and chalcopyrite were separated with a double flotation process in order to produce a pyrite and a chalcopyrite concentrate and relatively pyrite-poor tailings. After a break of 4 years, a single flotation replaced the double flotation process and, since then, pyrite has been reported to pyrite-rich tailings stored separately from the old ones. Both tailings were separately sampled: FA4 from the old pyrite-poor tailings, and FA3 from the current pyrite-rich tailings.

To reduce the AMD effect, we follow the traditional method of treating mine waters with calcium carbonate [30], to raise the pH up to the level of neutral or slightly acidic waters, such as the level of the meteoric ones,. This method involves adding the carbonate in sequential steps to the acid solution. The material used as buffer in this study is a fine-grained calcium carbonate provided by the Unicalce company, with specific characteristics (Table 2).

**Table 2.** Mineralogical and chemical composition of the buffering agent used in the tests. produced by the company Unicalce Spa.

| Unicalce S.p.A.-Technical Sheet |
|---|
| Product trade name: calcium carbonate paste<br>Chemical composition: |
| $CaCO_3$ (from CaO total) $\geq$ 90%<br>$MgCO_3$ (from MgO total) $\leq$ 4.0%<br>($SiO_2 + Al_2O_3 + Fe_2O_3$) $\leq$ 5.0%<br>$SO_3$ total: $\leq$ 0.25% |

Initially wet, the carbonate was dried in the oven at a temperature of about 70 °C until the complete removal of the wet component. The carbonate induces an increase in pH in the acid solution, that, in turn, decreases the solubility of ions in the solution, triggering their precipitation as amorphous material or hydroxide phases which incorporate the metal ions in their structure [31]. The final goal is the purification of the leached waters to ion contents lower than the legal limits for surficial waters.

The AMD evaluation of solid samples is based on the AMIRA procedure [9], which is a revision of the Sobek procedure [32]. The Acid-Base Account (ABA) tests are static laboratory procedures to evaluate the balance between acid-generating processes and acid-neutralising processes. The acid potential is referred to as the Maximum Potential Acidity (MPA), whereas the neutralizing potential is referred to as the Acid Neutralising Capacity (ANC). MPA is an estimate of the amount of acid that the sample can release by

complete oxidation of sulphides, expressed as kg $H_2SO_4$/t. The evaluation of MPA by the AMIRA standard procedure is based on the conservative assumption that all S is present as pyrite. This simplification may overestimate the AMD as other sulphides with higher Me/S ratio have lower acid generation potential than pyrite. Such an overestimation can give unrealistic results in the cases where high portions of S are present as non-acid generating phases (i.e., sulfates). ANC is an estimate of the buffering capacity of the sample expressed as kg $H_2SO_4$/t that the sample is able to neutralise. It is experimentally determined by titration preceded by a "fizz test" as described by Sobek et al. [32]. When a negative value of ANC is obtained, it is reported as 0.00, indicating the sample incapacity of neutralisation. The difference between MPA and ANC is referred to as the Net Acid Producing Potential (NAPP), which is used to indicate the acid-generating potential of a sample. The NAPP is also expressed in units of kg $H_2SO_4$/t. Negative values indicate that a sample may have sufficient ANC to prevent acid generation, whereas positive values indicate that a material may be acid-generating.

Buffering tests were performed on FA3 and FA4 leached samples, after separation between liquid and suspended/solid material by centrifuge, 4000 rpm for 4 min (Eppendorf Centrifuge 5702). For leaching, the procedure described by Hageman et al. [33] was followed.

Two buffering tests, short (S) and long (L), were conducted for each sample in order to collect precipitates at different pH values.

The quantities added in a 50 mL solution were in slots of 3, 5 and 10 mg, depending on the step test. The time required to reach the pH equilibrium in each buffering step was determined by kinetic chemical tests (Section 3.2.1). For each quantity added, the stirrer remained in operation for 30 min for the 3 mg/50 mL quantity and 60 min for the 5 mg/50 mL and 10 mg/50 mL quantities, respectively.

After leaching and buffering tests, the solutions were analyzed with two different techniques to evaluate the concentration of the same ions selected by Grieco et al. [16]. Sulfur percentage values were assumed based on the same study.

An inductively coupled plasma-mass spectroscopy (ICP-MS), Agilent 7500 ce System with 5-point calibration from 0 to 1 mg/L using ICP multi-element standard solution IV was used to determine Al, Co, Cu, Fe, Mn, Ni, Pb and Zn. An ion chromatograph Metrohm IC883+ equipped with autosampler 863 was used to determine Ca, Mg and $SO_4$. For chromatographic analysis samples were prepared by dilution in the ratios 1:10 for samples with EC of about 1 mS and 1:25 for samples with EC > 3 mS. Both ICP-MS and chromatographic analyses were performed at Department of Earth Sciences, University of Turin.

For a qualitative mineralogical analysis of precipitates via XRPD, after drying samples at 70 °C, a PANalytical X'Pert-pro was used, at the following operating conditions: 40 kV of voltage; 40 mA of current; Cu anticathode K$\alpha$1/K$\alpha$2: 1.540510/1.544330 Å. The data were elaborated with the software X'Pert Highscore v.2.3 (Malvern Panalytical, Malvern, UK).

Transmission electron microscopy (TEM) analysis was used to assess the presence of any nanocrystals or amorphous material into the precipitated materials: FEI Tecnai G2 F20 equipped with EDS microanalyses AztecEnergy, detector Xplore OXFORD at the Department of Earth Science, University of Milan, was used.

The $Fe^{2+}$/$Fe^{tot}$ ratio was determined both by ICP-MS analyses and by redox titration using $KMnO_4$, whereas $Fe^{3+}$ was calculated as the difference between the concentration of $Fe^{2+}$ and $Fe^{tot}$.

## 3. Modeling of Chemical Processes

In order to perform the buffering procedure, it is necessary to determine the optimal amount of $CaCO_3$ for neutralization. The minimization of the amount of buffering material has the double purpose of reducing the costs of the procedure and, at the same time, generating the lowest possible amount of sludge rich in PTE. For this purpose, we propose

a modeling of chemical processes by means of a numerical solution of the equilibrium equation, starting from the main ionic species present in the leached solution.

According to Grieco et al. [16] the cations present in high concentration and that require modeling are: $Fe^{2+}$, $Fe^{3+}$, $Al^{3+}$, $Cu^{2+}$, $Ca^{2+}$. Anions deriving from water equilibrium, dissolution of sulfides and carbonates and are: $SO_4^{2-}$, $HSO_4^-$, $CO_3^{2-}$, $HCO_3^-$, $OH^-$. The charge equilibrium equation can be then written as:

$$3\left[Fe^{3+}\right] + 2\left[Ca^{2+}\right] + [H^+] + 2\left[Fe^{2+}\right] + 3\left[Al^{3+}\right] + 2\left[Cu^{2+}\right]$$
$$= 2\left[SO_4^{2-}\right] + \left[HSO_4^-\right] + 2\left[CO_3^{2-}\right] + [HCO_3^-] + [OH^-] \tag{1}$$

The final purpose of the modeling is to find the relationship between the amount of $CaCO_3$ buffer added to the leached solution and the pH attained at charge equilibrium.

It is therefore essential to model the changes in the equilibrium due to the chemical reactions that are involved during $CaCO_3$ addition.

With the aim of obtaining a useful tool for the determination of the buffering settings, we built a theoretical model of these chemical reactions by numerical solution of the equilibrium equation, to define a buffering curve and determine the optimal amount of buffering material.

The acidification of water is intimately linked to the initial concentration of sulfuric acid present in the solution, in turn conditioned by the balance of the charges present in the solution. Pyrite is the main responsible for the acidification of water due to the following generic oxidation reaction (2) [34]:

$$FeS_2(s) + \frac{15}{4}O_2(l) + \frac{7}{2}H_2O = Fe(OH)_3(s) + 2\,SO_4^{2-}(l) + 4H^+ \tag{2}$$

By quantifying the acidity according to Equation (2), it is assumed that the initial pH is entirely due to the presence of $H_2SO_4$. Equation (2) basically shows that the sulfides, represented here in a simplified reaction by pyrite, generate acids and release sulfates by interaction with oxygen-rich waters.

Therefore, assuming that $H_2SO_4$ is the only acid present, the objective of this first part of the model is to determine the initial $H_2SO_4$ concentration required to bring an aqueous solution to the pH measured for our leached solutions. It is therefore possible to write the resulting equilibria:

$$H_2SO_4(l) \overset{strong}{\rightarrow} H^+(l) + HSO_4^-(l) \tag{3}$$

$$HSO_4^-(l) \overset{K_{as}}{\leftrightarrow} H^+(l) + SO_4^{2-}(l) \tag{4}$$

$$H_2O \overset{K_w}{\leftrightarrow} H^+(l) + OH^-(l) \tag{5}$$

Sulfuric acid is completely dissociated, according to Equation (2). The dissociation (4) is related to the dissociation constant $K_{as} = 1.02 \times 10^{-2}$. The third equilibrium (5) is instead determined by the normal dissociation of a water with $K_w = 1.00 \times 10^{-14}$. It is possible, starting from these equilibria, to write the following set of equations:

mass balance:

$$[H_2SO_4]_{init} = \left[HSO_4^-\right] + \left[SO_4^{2-}\right] + [H_2SO_4] \tag{6}$$

charge balance

$$[H^+] = [HSO_4^-] + 2\left[SO_4^{2-}\right] + [OH^-] \tag{7}$$

$HSO_4^-$ equilibrium

$$K_{as} = \frac{[H^+]\left[SO_4^{2-}\right]}{[HSO_4^-]} \tag{8}$$

$H_2O$ equilibrium

$$K_w = [H^+][OH^-] \tag{9}$$

The only known parameter is the concentration of $H^+$ in the solution:

$$[H^+] = 10^{-pH} \frac{mol}{l} \tag{10}$$

The system is composed of four equations and four unknowns, and is therefore analytically solvable. After some simple steps, we achieve:

$$[HSO_4^-] = \frac{[H^+][H_2SO_4]_{init}}{K_{as} + [H^+]} \tag{11}$$

$$[SO_4^{2-}]_{th} = [H_2SO_4]_{init} \cdot \left(1 - \frac{[H^+]}{K_{as} + [H^+]}\right) \tag{12}$$

The charge balance equation can be written as:

$$[H^+] = \frac{K_w}{[H^+]} + \frac{[H^+][H_2SO_4]_{init}}{K_{as} + [H^+]} + 2[H_2SO_4]_{init} \cdot \left(1 - \frac{[H^+]}{K_{as} + [H^+]}\right) \tag{13}$$

Since $K_w$ and $K_{as}$ are known, $[H_2SO_4]_{init}$ depends solely on $[H^+]$. Solving the calculations and defining the Equation (13) with respect to $[H_2SO_4]_{init}$, we achieve:

$$[H_2SO_4]_{init} = \frac{[H^+]^2 \cdot (K_{as} + [H^+]) - K_w \cdot K_{as} - K_w + [H^+]}{2[H^+] \cdot (K_{as} + [H^+]) - [H^+]^2} \tag{14}$$

### 3.1. Modeling of Soluble Ions

As the leaching waters contain other soluble ionic species besides those appearing in Equation (6) [16], these have to be considered as well. The major ions selected for the implemented model according to their concentration in leaching waters are: $[Fe^{2+}]$ $[Fe^{3+}]$ $[Al^{3+}]$ $[Cu^{2+}]$ $[Ca^{2+}]$. Each ion dissolved in the leachates is responsible for a change in the charge balance and, therefore, also in $[H_2SO_4]_{init}$.

In order to quantify the effects of the ions in the solution, it was necessary to insert in reaction (2) the concentrations in mol/l of all the ions present, taking into account their positive/negative charges. Furthermore, the solubility of a generic ion is heavily affected by the pH of the solution, and, during the buffering, it is possible that precipitation occurs, giving rise to changes in the acidity of the solution. In order to determine the correlation between pH of the leachates and the amount of buffering agent added, it was therefore necessary to integrate the effect of the ions on the charge balance with the variation of the solubility of each ion during buffering.

### 3.1.1. Modeling of $Al^{3+}$, $Fe^{2+}$, $Fe^{3+}$, $Cu^{2+}$

In this paragraph, we analyze in detail the modeled behavior of $Al^{3+}$, assuming that the other three major ions selected ($Fe^{2+}$, $Fe^{3+}$, $Cu^{2+}$) follow the same path. $Ca^{2+}$, as explained in Section 3.1.2, follows a different modeling approach.

The first step consists of inserting the chosen ion in the charge balance. Consider the following chemical balance:

$$Al(OH)_3(s) \overset{K_{ps}}{\leftrightarrow} Al^{3+}(l) + 3OH^-(l) \tag{15}$$

whose solubility product is

$$K_{ps} = [Al^{3+}][OH^-]^3 \tag{16}$$

Writing the $OH^-$ as a function of the dissociation constant of the water and explaining with respect to the $Al^{3+}$, we achieve:

$$\left[Al^{3+}\right] = K_{ps} \cdot \frac{\left[H^+\right]^3}{K_w{}^3} \tag{17}$$

where $\left[Al^{3+}\right]$ is the maximum amount of soluble aluminium ions. The solubility product of aluminium hydroxide is $K_{ps} = 1.90 \times 10^{-32}$. To reflect the effects of the $Al^{3+}$ on the acidification of the leached solution, it is necessary to quantify the concentration of the same ion in the solution at each pH. It is possible to mathematically write this last statement as:

$$\text{if } \left[Al^{3+}\right]_{\text{measured}} < \text{solublility} \Rightarrow \left[Al^{3+}\right]_{\text{in solution}} = \left[Al^{3+}\right]_{\text{measured}}$$

$$\text{if } \left[Al^{3+}\right]_{\text{measured}} > \text{solublility} \Rightarrow \left[Al^{3+}\right]_{\text{in solution}} = \text{solublility}$$

Once the concentration value of the $Al^{3+}$ in solution has been determined, it can be inserted into the charge balance Equation (1) as $3\left[Al^{3+}\right]$. For $Fe^{2+}$, $Fe^{3+}$, $Cu^{2+}$, we have to add $2\left[Fe^{2+}\right]$, $3\left[Fe^{3+}\right]$ and $2\left[Cu^{2+}\right]$ to Equation (1).

3.1.2. Modeling of Soluble $Ca^{2+}$

$Ca^{2+}$ requires a special precaution in modeling the solubility/precipitation processes as it can be present both as sulphate and as carbonate. We must consider that the amount of $Ca^{2+}$; therefore, its solubility equilibrium depends both on the $Ca^{2+}$ already present in the solution and on the $Ca^{2+}$ introduced with the $CaCO_3$ buffer.

Precipitation of Carbonates

$Ca^{2+}$ equilibrium is defined by a system of three equations in three unknowns as follows:

$$\begin{cases} [Ca^{2+}] = [H_2CO_3] + [HCO_3^-] + \left[CO_3^{2-}\right] \\ K_{a1} = \frac{[H^+]\ [HCO_3^-]}{[H_2CO_3]} \\ K_{a2} = \frac{[H^+]\ [CO_3^{2-}]}{[HCO_3^-]} \\ [Ca^{2+}] = \frac{K_{ps}}{[CO_3^{2-}]} \end{cases} \tag{18}$$

whose explicit result as a function of $\left[Ca^{2+}\right]$ is:

$$\left[Ca^{2+}\right] = \sqrt{K_{ps} \cdot \left(1 + \frac{\left[H^+\right]^2}{K_{a1} \cdot K_{a2}} + \frac{\left[H^+\right]}{K_{a2}}\right)} \tag{19}$$

where $\left[Ca^{2+}\right]$ is the maximum amount of the soluble calcium ion. Constants have the following values: $K_{ps} = 8.7 \times 10^{-9}$, $K_{a1} = 4.3 \times 10^{-7}$ and $K_{a2} = 5.6 \times 10^{-11}$. To reflect the effects of $Ca^{2+}$ on the acidification of the leached solution, it is necessary to quantify the concentration of the same ion in solution at each pH. It is possible to mathematically write this last statement as:

$$\text{if } \left[Ca^{2+}\right]_{\text{measured}} < \text{solublility} \Rightarrow \left[Ca^{2+}\right]_{\text{in solution}} = \left[Ca^{2+}\right]_{\text{measured}}$$

$$\text{if } \left[Ca^{2+}\right]_{\text{measured}} > \text{solublility} \Rightarrow \left[Ca^{2+}\right]_{\text{in solution}} = \text{solublility}$$

where $\left[Ca^{2+}\right]_{measured}$ is the amount present in the solution.

Once the concentration value of the $Ca^{2+}$ in solution has been found, it is possible to insert it in the charge balance shown in Equation (1) as $2\left[Ca^{2+}\right]$.

### 3.1.3. Precipitation of Sulfates

The modeling approach for the precipitation of sulfates is different from that used for carbonates due to the presence of $SO_4^{2-}$ in the initial solution.

The only relevant sulfate that can precipitate at the pH range of interest is $CaSO_4$, whose equilibrium is:

$$CaSO_4(s) \overset{K_{ps}}{\Leftrightarrow} Ca^{2+}(l) + SO_4^{2-}(l) \tag{20}$$

and whose solubility product assumes the inequality relationship:

$$\left[Ca^{2+}\right]\left[SO_4^{2-}\right] \leq K_{ps} = 6.85 \times 10^{-5} \tag{21}$$

where the equal sign indicates saturation, while the minor sign is for the under-saturated solution. The theoretical product of $\left[Ca^{2+}\right]\left[SO_4^{2-}\right]$ is numerically calculated, because $\left[Ca^{2+}\right]$ is known by (19) and $\left[SO_4^{2-}\right]$ is also known, from the acidity initial solution.

The inequality (21) must always be satisfied. If the result of the theoretical calculation of this product is greater than $K_{ps}$ there will be precipitation of $CaSO_4$, shifting the equilibrium to the left:

$$\left[Ca^{2+}\right]_{th} = \begin{cases} [Dose] + \left[Ca^{2+}\right]_{measured} & \text{if } [Dose] + \left[Ca^{2+}\right]_{measured} < \text{solubility} \\ \text{solubility if } [Dose] + \left[Ca^{2+}\right]_{measured} > \text{solubility} \end{cases} \tag{22}$$

where $\left[Ca^{2+}\right]_{th}$ is the total amount in solution before buffering, "Dose" is the amount of buffer added, as $CaCO_3$, and $\left[SO_4^{2-}\right]_{th}$ the sum of $[H_2SO_4]_{init}$ plus the quota precipitated (12). Calling x the number of moles of $Ca^{2+}$ precipitating, we can rewrite the (21) as:

$$\left(\left[Ca^{2+}\right]_{th} - x\right) \cdot \left(\left[SO_4^{2-}\right]_{th} - x\right) = K_{ps} \tag{23}$$

After having solved (23), with respect to the unknown x, it is possible to calculate the real concentrations of the ions at equilibrium as a function of pH, to be included in the overall charge balance (1) as $2\left[Ca^{2+}\right]$ and $2\left[SO_4^{2-}\right]$. The following equations summarize the values of $Ca^{2+}$ and $SO_4^{2-}$:

$$\left[Ca^{2+}\right] = \begin{cases} \left[Ca^{2+}\right]_{th} & \text{if } \left[Ca^{2+}\right]_{th} \cdot \left[SO_4^{2-}\right]_{th} \leq K_{ps} \\ \left[Ca^{2+}\right]_{th} - x & \text{if } \left[Ca^{2+}\right]_{th} \cdot \left[SO_4^{2-}\right]_{th} > K_{ps} \end{cases} \tag{24}$$

$$\left[SO_4^{2-}\right] = \begin{cases} \left[SO_4^{2-}\right]_{th} & \text{if } \left[Ca^{2+}\right]_{th} \cdot \left[SO_4^{2-}\right]_{th} \leq K_{ps} \\ \left[SO_4^{2-}\right]_{th} - x & \text{if } \left[Ca^{2+}\right]_{th} \cdot \left[SO_4^{2-}\right]_{th} > K_{ps} \end{cases} \tag{25}$$

The pH dependence of these equations is not explicit. For calcium, the dependence is within the value of $\left[Ca^{2+}\right]_{th}$, the calculation of which comes from (19), where the dependence on pH is explicit. For the sulphate anion, the dependence on pH is more evident, and follows Equation (8). Finally, it is important to note how all the functions that regulate the number of ions in the solution, as a result of the precipitation, are defined by sections, thus complicating the solution of the charge balance as shown in Equation (1).

*3.2. Buffering: Theoretical Determination of the Optimal Amount of Buffering Material*

The modeling of $[H_2SO_4]_{init}$ necessary to bring an aqueous solution to a given pH is now complete, in equilibrium with all the major ions present in the leached solution. It is therefore possible to define the variation of pH as a function of the amount of $CaCO_3$ added to the leachates as a buffer. To determine the relationship between $CaCO_3$ added as buffer and the final pH, it is necessary to solve the charge equilibrium equation:

$$3\left[Fe^{3+}\right] + 2\left[Ca^{2+}\right] + [H^+] + 2\left[Fe^{2+}\right] + 3\left[Al^{3+}\right] + 2\left[Cu^{2+}\right]$$
$$= 2\left[SO_4^{2-}\right] + \left[HSO_4^-\right] + 2\left[CO_3^{2-}\right] + [HCO_3^-] + [OH^-] \tag{26}$$

where $[HCO_3^-]$ can be written as a function of $[H^+]$ following the calcium carbonate equilibrium as:

$$[HCO_3^-] = [CaCO_3]_{init} \cdot \left(1 + \frac{[H^+]}{K_{a1}} + \frac{K_{a2}}{[H^+]}\right)^{-1} \tag{27}$$

Despite the fact that the only unknown parameter is $[H^+]$, (26) is a non-linear piecewise analytically unsolvable equation. It is therefore necessary to solve this equation numerically using an Excel sheet with the following constraints:

i.     Discretization of pH with 0.01 steps;
ii.    Calculation of each ion for each discretized pH;
iii.   Sum pos = sum neg;
iv.    Resolution of the following minimum problem:

$$[H^+] = \min_{pH} \left(\sum_i n_i \left[P^{n_i+}\right]_i - \sum_j m_j \left[N^{m_j-}\right]_j\right)^2 \tag{28}$$

where $[p^{n_i+}]_i$ indicates the generic positive ion P of valence $n_i$ and $\left[N^{m_j-}\right]_j$ indicates the generic negative ion N of valence $m_i$. The range of values over which the minimum is sought is defined by the $[H^+]$ limits: $[10^{-14};10^{-1}]$. The solution of the equation is the pH which minimizes the squared difference between the sum of the positive charges and the sum of the negative charges, and which therefore best approximates the equation with the discretization step chosen.

3.2.1. Buffering: Determination of the Time Required to Attain Equilibrium

To ensure that the metal ions precipitate in secondary phases, it is necessary that the neutralization reactions take place completely. For these reactions (the most common is the precipitation of hydroxides) the precipitation rate is rather slow. Therefore, it was decided to conduct preliminary tests, adding a small amount of buffering material by steps, encouraging a faster reaction with a stirrer and measuring the pH for each step. The best configuration obtained has been 10 mg/50 mL per hour (Figure 2).

3.2.2. $Fe^{2+}/Fe^{tot}$ Ratio Analysis in Solution

The $Fe^{2+}/Fe^{tot}$ ratio was identified by ICP-MS and tritration, which yielded consistent results. It can be observed that the titrations show that ferrous ion concentrations are slightly higher than those of total iron. This small difference, however, can be explained by the different sensitivity of the two techniques and, therefore, it is possible to assume that the iron present in solution is mostly in the form of ferrous ion. For the buffering modeling, it assumed that $Fe^{2+}/Fe^{tot}$ is 95%.

Regarding the analysis by titration, potassium permanganate ($KMnO_4$) is used as an oxidising agent. It follows the chemical reaction for iron oxidation (29):

$$MnO_4^-(l) + 5\,Fe^{2+}(l) + 8\,H^+(l) \rightarrow Mn^{2+}(l) + 5\,Fe^{3+}(l) + 4\,H_2O \tag{29}$$

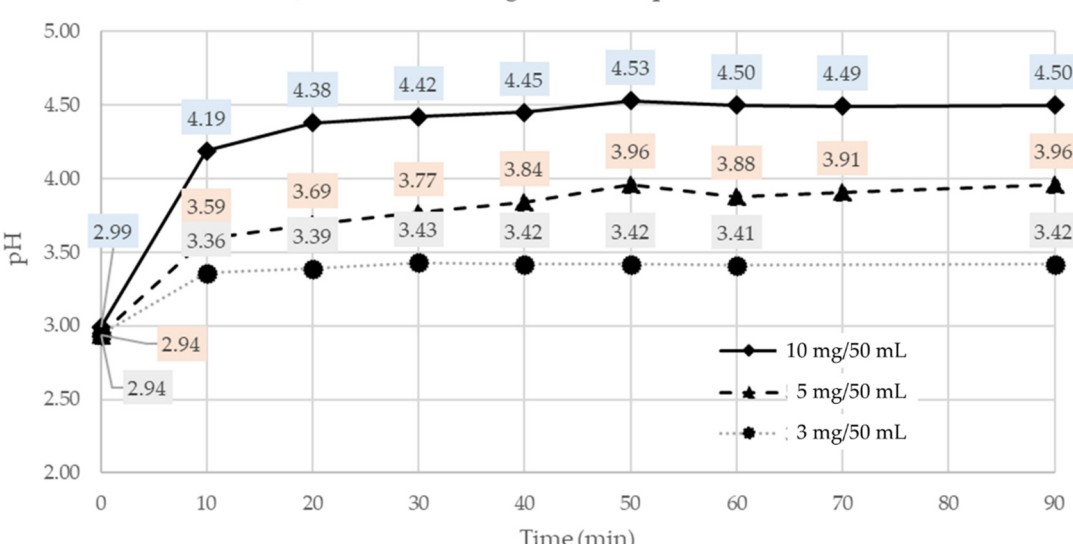

**Figure 2.** Result of reaction time to obtain the equilibrium, between samples and buffering carbonate added. After one hour, the pH value has stabilized.

## 4. Results

ABA and buffering tests and chromatography, ICP-(MS), XRD and TEM analysis are used to determine AMD and related pH values before/after buffering, considering concentrations of ions in solutions and mineral phases of precipitates. All results are partially compared with analyses reported by Grieco et al. [16].

### 4.1. ABA Tests

ABA test results are reported in Table 3. According to Price (1997), the criteria for evaluating AMD through NAPP are as follows: if the NAPP is higher than 20 kg $H_2SO_4/t$, it is generally accepted that the material is not acid producing; if the NAPP is lower than $-20$ kg $H_2SO_4/t$, it is generally accepted that the material is acid-producing; values in the range $-20 < NAPP < +20$ kg $H_2SO_4/t$ are uncertain, and kinetic tests may be needed.

**Table 3.** Calculation of ANC ANC = [Y × $M_{HCl}$/wt] × C, where: Y = (Vol. of HCl added) − (Vol. of NaOH titrated × B); B = (Vol. of HCl in blank)/(Vol. of NaOH titrated in blank); C = Conversion factor (to calculate kg $H_2SO_4/t$).

| Sample | FA3 | FA4 |
|---|---|---|
| Vol (mL) HCl sample | 20.0 | 20.0 |
| Vol (mL) NaOH sample | 29.7 | 23.0 |
| Vol (mL) HCl blank | 20.0 | 20.0 |
| Vol (mL) NaOH blank | 26.5 | 22.1 |
| Weight (g) sample | 2.0 | 2.0 |
| $S_{tot}$ wt% | 36.8 | 15.9 |
| Fizz Rating | 3 | 3 |
| Molarity (M) of HCl | 0.5 | 0.5 |
| Molarity (M) of NaOH | 0.5 | 0.5 |
| Y | 0 (−2.41) | 0 (−0.81) |
| B | 0.75 | 0.90 |
| C | 49 | 49 |
| ANC (kg $H_2SO_4/t$) | 0 (−29.3) | 0 (−9.97) |
| MPA (kg $H_2SO_4/t$) | 1150 | 497 |
| NAPP (kg $H_2SO_4/t$) | 1150 | 497 |

The FA3 sample has the highest sulfur content (36.8 wt%), which is responsible for a high acid drainage potential (NAPP = 1150 kg $H_2SO_4/t$).

The FA4 sample has a S content two times lower (15.9 wt%) than FA3. Acid drainage potentials are lower than FA3, but still remain high (NAPP = 497 kg $H_2SO_4/t$).

Both samples have no buffering potential; hence, the NAPP value coincides with the MPA one.

### 4.2. Leaching and Buffering Tests: Water Analysis (Chromatography, ICP-MS)

FA3 and FA4 samples produce, as a result of leaching tests, strongly acidic (2.99 < pH < 3.51) and metal-rich solutions (Table 4).

**Table 4.** Chemical–physical parameters and metal contents in leachates, before buffering (FA3 and FA4 liquid samples) and in short and long buffered solutions (FA3-S, FA3-L, FA4-S and FA4-L liquid samples). Two different methods of metal content analysis were used, ICP-MS (indicated with *) and chromatography (indicated with **).

| Sample | | Leached FA3 | Buffered FA3-S | FA3-L | Leached FA4 | Buffered FA4-S | FA4-L |
|---|---|---|---|---|---|---|---|
| water acidity | pH | 2.99 | 5.01 | 6.62 | 3.51 | 5.62 | 7.41 |
| Conductivity | μS/cm | 3880 | 3820 | 3870 | 1224 | 1305 | 1396 |
| Temperature | °C | 22.7 | 22.7 | 22.7 | 22.7 | 22.7 | 22.7 |
| Al * | mg/L | 159 | 5.55 | 0.29 | 83.3 | 0.43 | 0.3 |
| As * | μg/L | 13 | 1.32 | n.d. | 7.4 | 0.64 | n.d. |
| Ca ** | mg/L | 519 | 768 | 827 | 37.6 | 221 | 238 |
| Co * | mg/L | 0.55 | 0.62 | n.d. | 0.41 | 0.44 | 0.1 |
| Cu * | mg/L | 14.9 | 10.6 | n.d. | 10.7 | 0.93 | 0.07 |
| Fe * | mg/L | 391 | 224 | 70.9 | 16.7 | 3.28 | 2.54 |
| Mg ** | mg/L | 131 | 137 | 56.4 | 58.8 | 58.7 | 56.3 |
| Mn * | mg/L | 9.72 | 10.9 | 9.68 | 1.55 | 1.62 | 1.31 |
| Ni * | mg/L | n.d. | n.d. | n.d. | n.d. | n.d. | n.d. |
| Pb * | mg/L | n.d. | n.d. | n.d. | n.d. | n.d. | n.d. |
| $SO_4^{2-}$ ** | mg/L | 2589 | 2469 | 2248 | 676 | 806 | 816 |
| Zn * | mg/L | 2.91 | 2.98 | 0.11 | 0.91 | 0.74 | n.d. |

Some chemical species show strong variations, comparing the initial concentrations of leachates with their amount during buffering. Increasing pH gradually produces new chemical equilibria that change the initial composition of solutions.

For major elements within sample FA3, the most sensitive changes in ion concentrations occur after short buffering (final pH = 5.01) and are given by: Al (−97%), Ca (+48), Cu (−29%) and Fe (−43%). Likewise, for major elements, the FA4 sample shows the highest variability, after short buffering, for: Al (−99%), Ca (+476%), Cu (−44%) and Fe (−80%).

Buffering tests were conducted by steps of addition of buffer agent (Table 5):

- To evaluate pH variation vs. amount buffered added;
- To validate the geochemical model.

During the buffering tests, it was decided to divide each FA3 and FA4 sample, respectively into FA3-S (short), FA3-L (long), FA4-S and FA4-L subgroups, in order to:

- Recover and analyse the ions into the solution at intermediate steps;
- Recover and analyse the precipitates at intermediate steps.

For all samples, as the acidity of the solution increases, the conductivity decreases till pH = 4, and then rises back to approximately the initial values. With the same buffering agent concentration (1584 mg/L) present in the solutions, sample FA3 showed the lowest increase in acidity with pH = 5.34, whereas sample FA4 reached pH = 7.41.

**Table 5.** Variation of acidity and conductivity during buffering step on samples FA3-S, FA3-L, FA4-S and FA4-L as a function of the buffer amount added.

| Buffer | | FA3-S | | FA3-L | | FA4-S | | FA4-L | |
|---|---|---|---|---|---|---|---|---|---|
| Added (mg) | Concentra-tion (mg/L) | pH | mS/cm | pH | mS/cm | pH | mS/cm | pH | mS/cm |
| 0 | 0 | 2.99 | 3.88 | 2.99 | 3.88 | 3.51 | 1.22 | 3.70 | 1.22 |
| 3 | 60 | 3.33 | 3.70 | 3.33 | 3.74 | 3.99 | 1.16 | 3.90 | 1.19 |
| 3 | 120 | 3.66 | 3.66 | 3.69 | 3.67 | 4.13 | 1.17 | 4.08 | 1.19 |
| 5 | 240 | 4.03 | 3.63 | 4.05 | 3.65 | 4.20 | 1.19 | 4.16 | 1.21 |
| 5 | 360 | 4.13 | 3.52 | 4.16 | 3.65 | 4.29 | 1.22 | 4.24 | 1.24 |
| 10 | 560 | 4.21 | 3.63 | 4.24 | 3.69 | 4.64 | 1.26 | 4.45 | 1.28 |
| 10 | 760 | 4.35 | 3.69 | 4.37 | 3.73 | 5.62 | 1.30 | 5.58 | 1.34 |
| 10 | 960 | 4.53 | 3.77 | 4.52 | 3.74 | | | 6.50 | 1.37 |
| 10 | 1160 | 4.68 | 3.81 | 4.65 | 3.79 | | | 6.66 | 1.39 |
| 10 | 1360 | 5.01 | 3.82 | 5.15 | 3.84 | | | 7.41 | 1.40 |
| 10 | 1560 | | | 5.58 | 3.87 | | | | |
| 10 | 1760 | | | 5.77 | 3.89 | | | | |
| 10 | 1960 | | | 6.11 | 3.93 | | | | |
| 10 | 2160 | | | 6.44 | 3.91 | | | | |
| 10 | 2360 | | | 6.62 | 3.90 | | | | |

### 4.3. Precipitates Mineralogy (XRPD, TEM)

During the buffering session, precipitates were formed due the pH and chemical equilibrium variation because of the reaction between buffering agent and solution. For all buffered samples (FA3-S, FA3-L, FA4-S, FA4-L) precipitated mineralogical phases were collected and investigated by XRPD (Figures 3–5) and they are characterized by the presence of ubiquitous carbonates (dominant calcite and subordinate dolomite), hydro-silicates of varying composition, hydroxides mainly of aluminum, sulfates and a minor amount of quartz.

Precipitates of the FA3-S buffered sample (Figure 3) show the presence of Ca-Fe-sulfates, Ca-Fe-silicates, carbonates, Al-hydroxides and quartz, whereas the same sample at higher pH (FA3-L) (Figure 4) mainly lacks silicate phases.

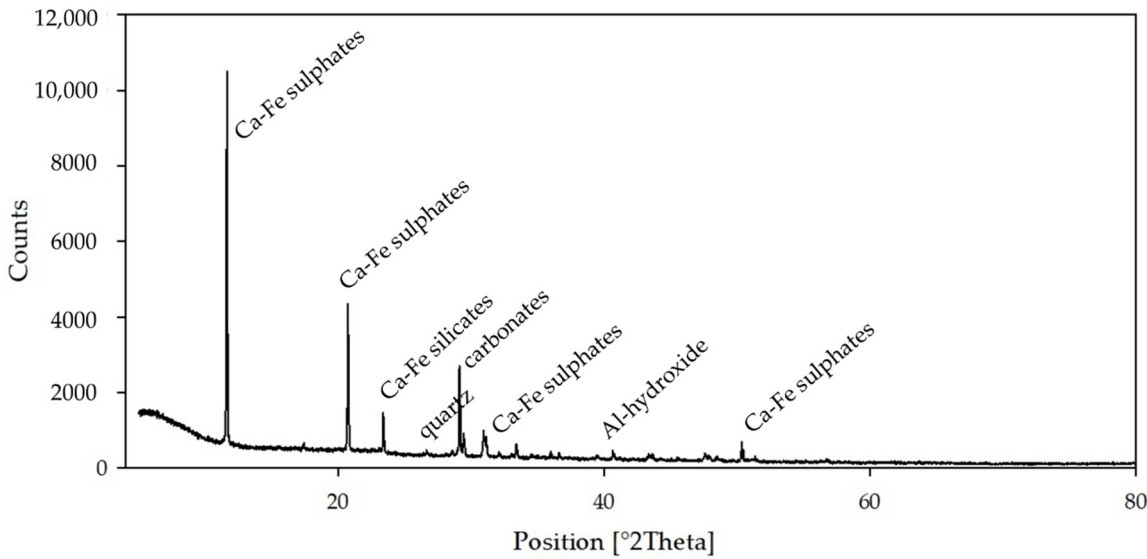

**Figure 3.** XRPD pattern of FA3-S precipitates and qualitative analyses for content of mineral phases.

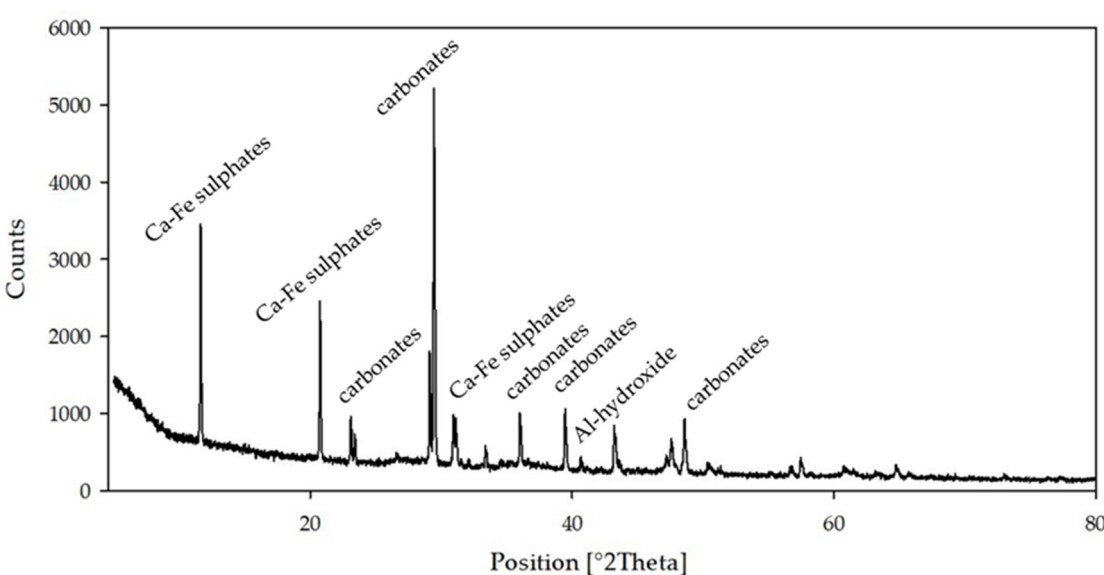

**Figure 4.** XRPD pattern of FA3-L precipitates and qualitative analyses for content of mineral phases.

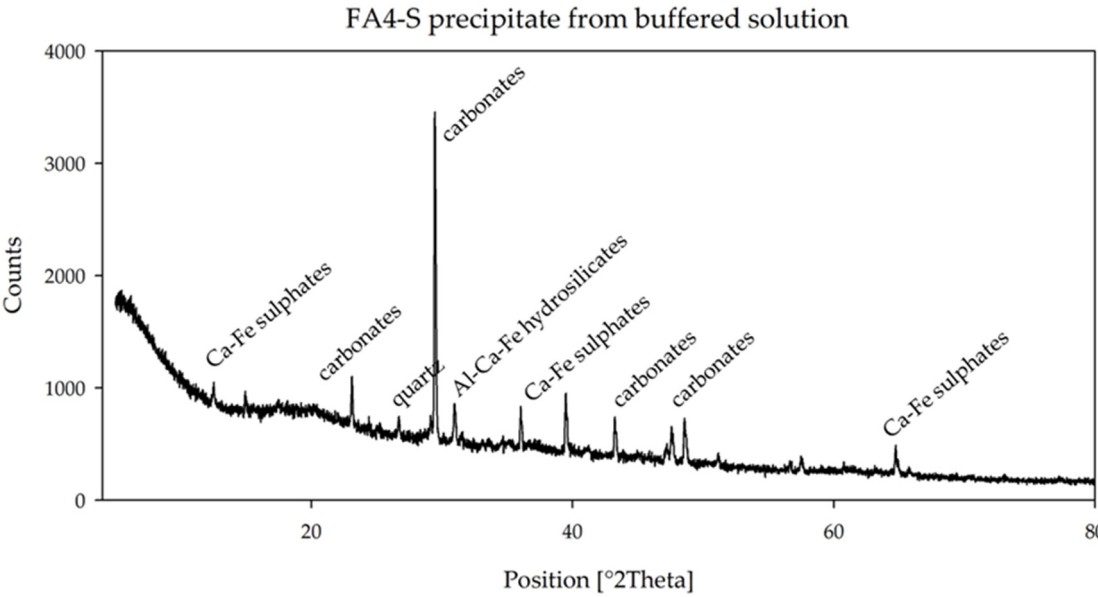

**Figure 5.** XRPD pattern of FA4-S precipitates and qualitative analyses for content of mineralogical phases.

Precipitates of sample FA4-S show the presence of Ca-Fe-sulfates, Al-Ca-Fe-hydrosilicates, carbonates and quartz (Figure 5), while precipitates at higher pH (sample FA4-L) (Figure 6), are mainly composed of carbonates.

Interpretation of the powder diffraction patterns suggests the possible presence of nanocrystalline or amorphous material. For this purpose and as a preliminary step, a TEM analysis was performed only for the FA3-S leachate precipitate. Together with the aid of EDS instrumentation, the results of the analysis provided valuable information on the size, shape, structure, possible aggregation state of the particles, and chemistry of the observed phases. Thus, images of phases or their aggregates, diffraction patterns (if any), and EDS spectra with the detected chemical species, such as Al, Ca, Fe, Mg, and S (Figures 7–10), were collected for each main mineralogical species identified: Ca-Fe-sulfates, Al-Ca-Fe-silicates, carbonates, Al-hydroxides.

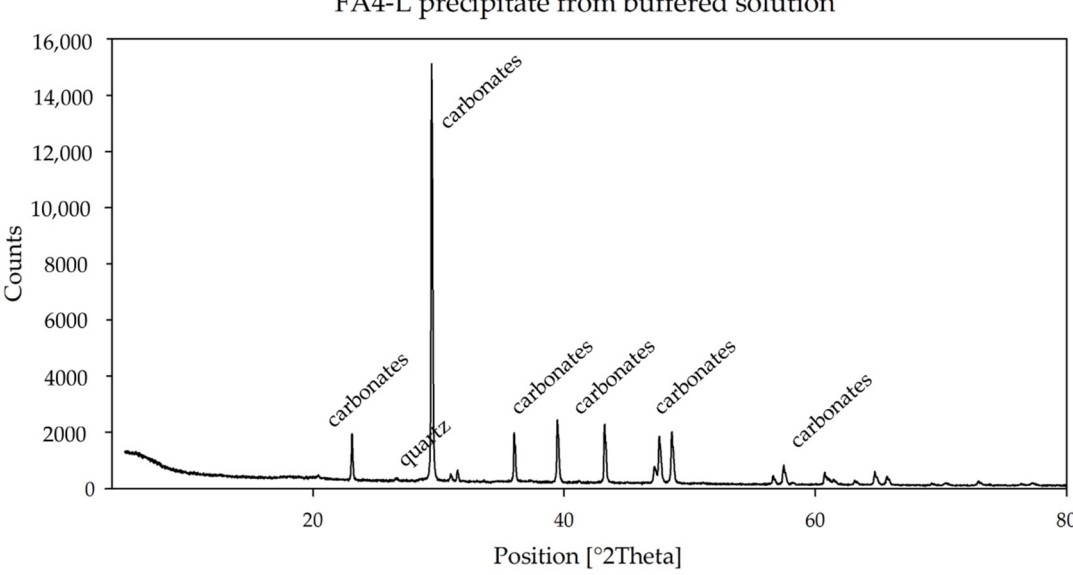

**Figure 6.** XRPD pattern of FA4-L precipitates and qualitative analyses for content of mineralogical phases.

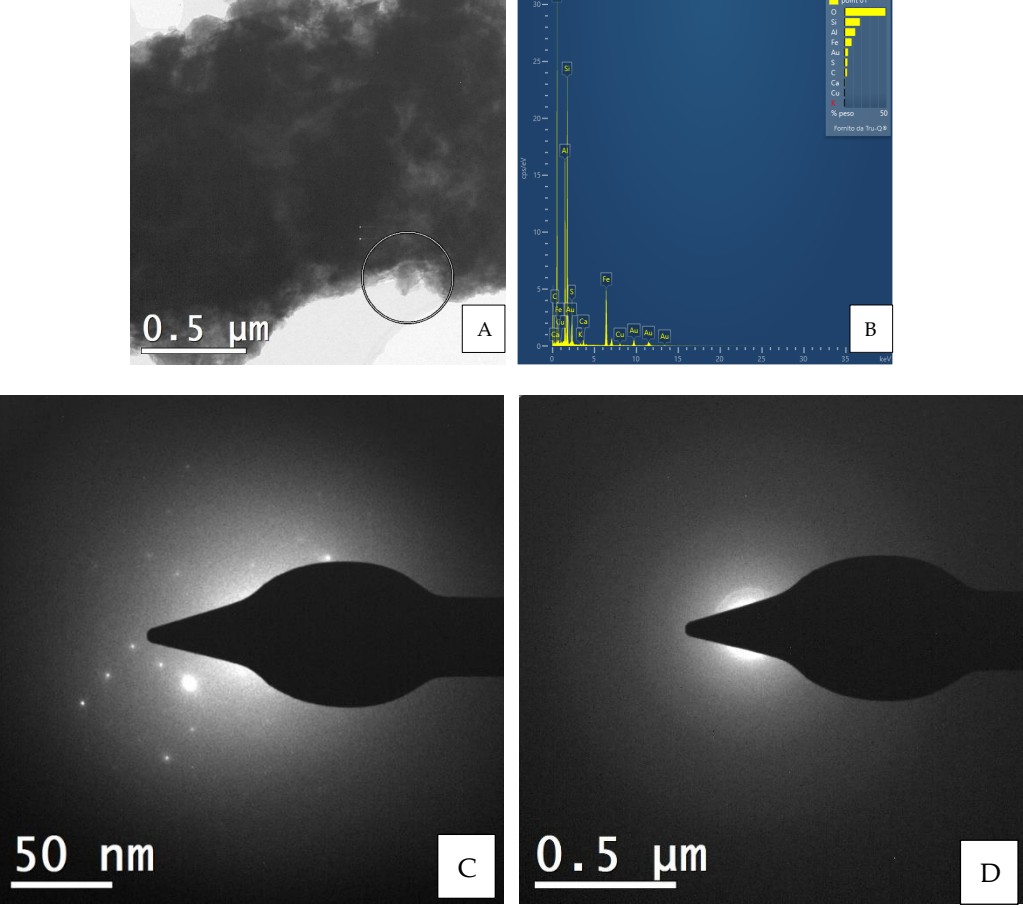

**Figure 7.** Precipitated phases from FA3-S sample, analyzed by TEM. A nano-crystals aggregate (**A**) probably an Al-Fe-(hydro)silicate phase, as suggested by EDS analysis (**B**). The single-crystal pattern (**C**) shows an evident crystalline structure before the interaction with electron beam of the EDS analysis. After EDS beam there is no pattern (**D**), because the structure has been dissolved. The black figure in the center of the images (**C**) and (**D**), whose size remains unchanged as the magnification changes, is the beam stopper.

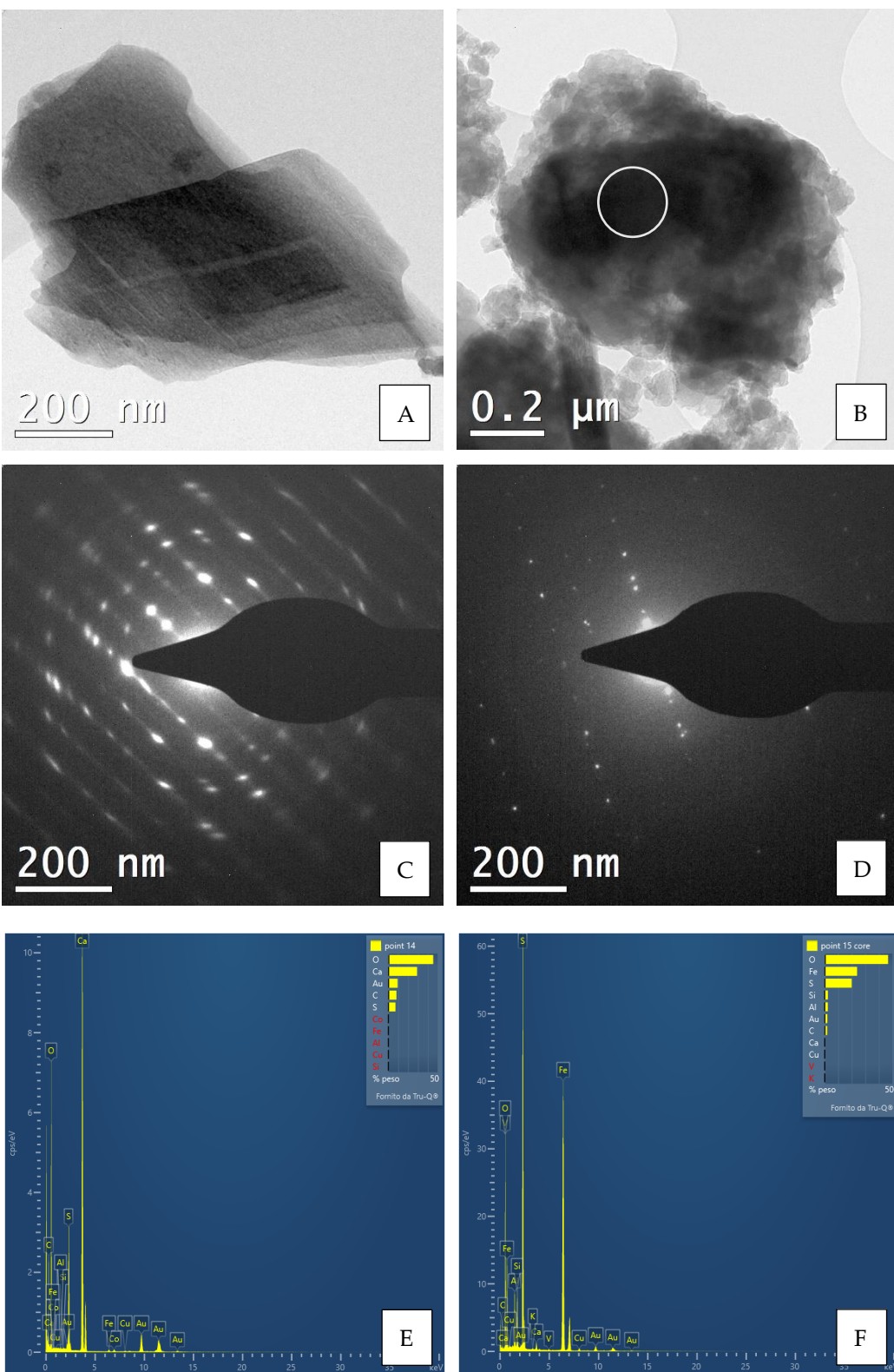

**Figure 8.** Precipitated phases from FA3-S sample, analyzed by TEM. Two nanocrystal, as the the diffraction patterns reveal: on the left column (**A**,**C**,**E**), eudral nanocrystals of Ca-sulfate; on the right side (**B**,**D**,**F**), probably a Fe-sulfate nanocrystal covered by aggregates of the amorphous phase.

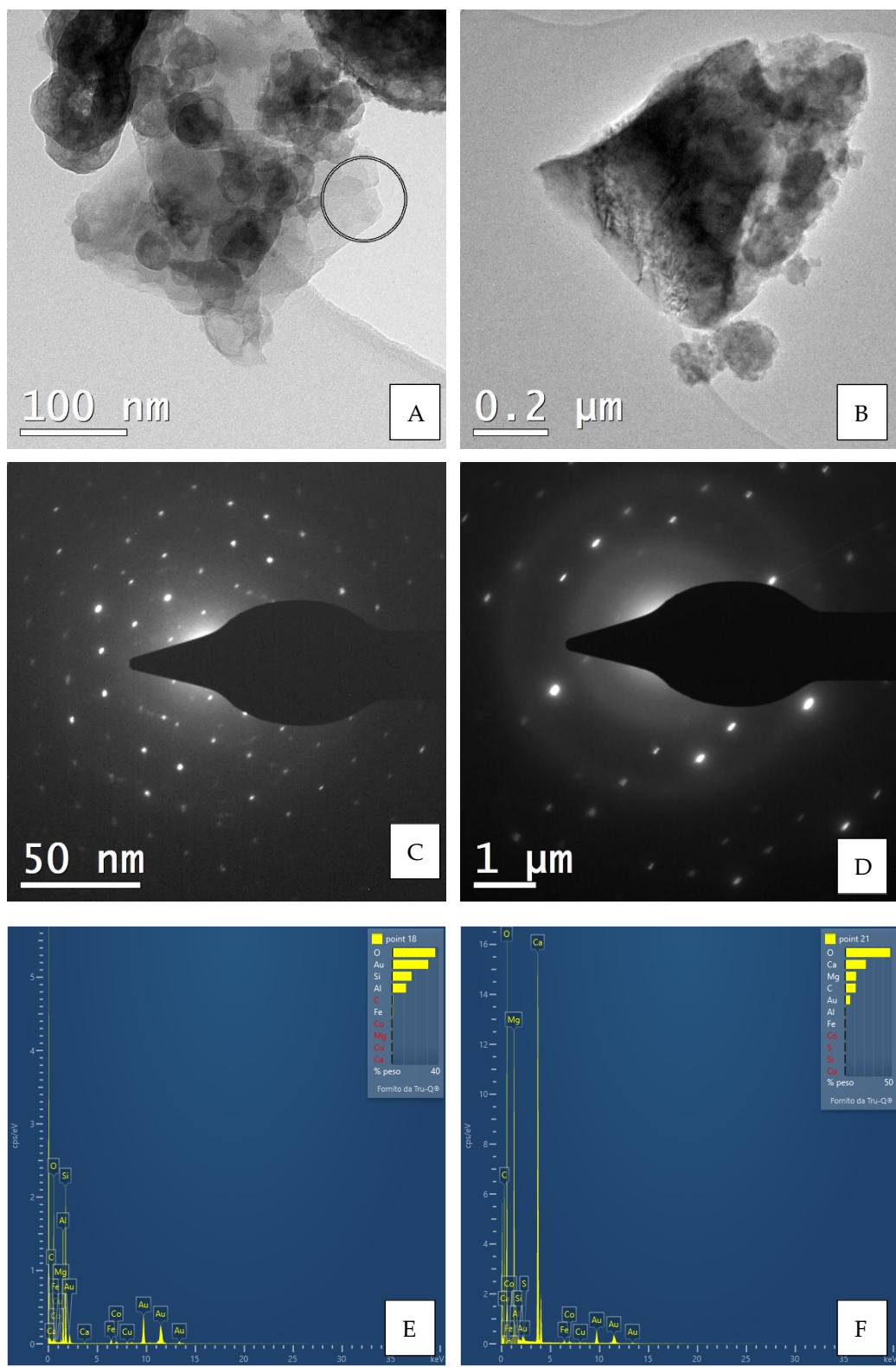

**Figure 9.** Precipitated phases from FA3-S sample, analyzed by TEM. Nano-crystal aggregates (**A**) and subhedral crystal (**B**) with relative diffraction patterns (**C**,**D**). Their EDS analyses show an Al-silicate (hydrated?) (**E**) and a probable Mg-carbonate (**F**).

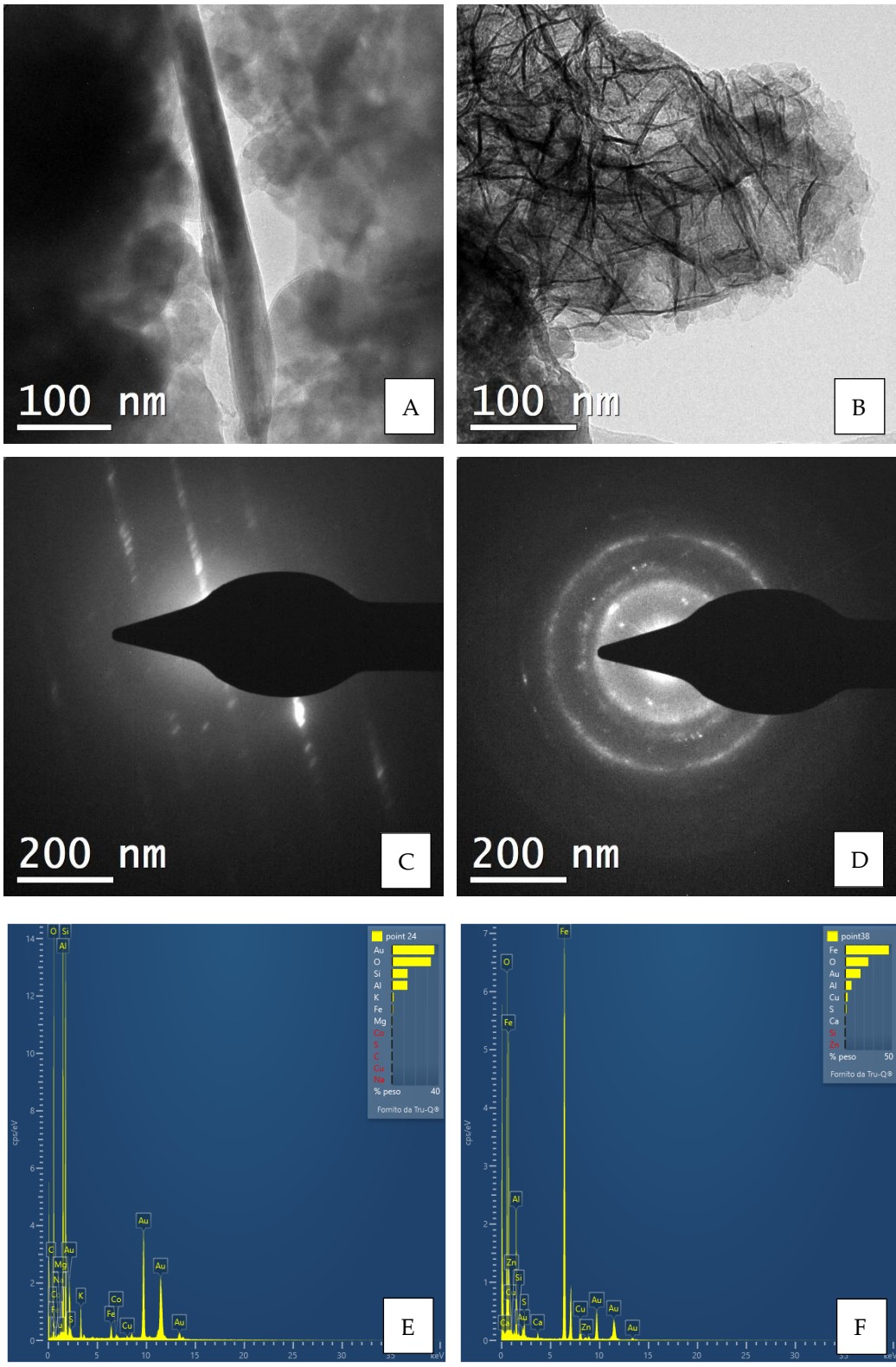

**Figure 10.** Precipitated phases from FA3-S sample, analyzed by TEM. Nano-acicular crystal (**A**) and crystalline aggregate (**B**) with relative diffraction patterns (**C**,**D**). Their EDS analyses show an Al-K-(hydro)silicate (**E**) and a Fe-(hydro)oxide (**F**) phases.

## 5. Discussion

### 5.1. Verification of Geochemical Modeling

The geochemical model of the equilibrium reactions and the results of the buffering tests presented in the previous chapters can now be compared. The following graphs (Figures 11 and 12) represent the comparison between the pattern calculated with the geochemical model and the observed pattern resulting from the buffering tests for samples FA3 and FA4, respectively.

In Figure 11, referring to sample FA3, the theoretical curve shows the trend of pH change as a function of the buffering material added as predicted by the model. The acidity of the solution shows nearly stable values for buffer concentration between 200 mg/L and 1200 mg/L and after 1600 mg/L. In these ranges, the lack of pH change implies the formation of precipitates [35,36].

The red diamonds, on the other hand, correspond to the amounts of buffering agent added during the test and, again, we can observe a range of buffer concentration values, between 240 mg/L and 1160 mg/L, with minimal pH change. Thereafter, no other flat pH trend occurs, but it increases slowly.

The fit between calculated and observed data is almost ideal until pH = 4.63, whereas it is slightly different after pH = 5. From an experimental point of view, the overall result can be considered to be definitely good.

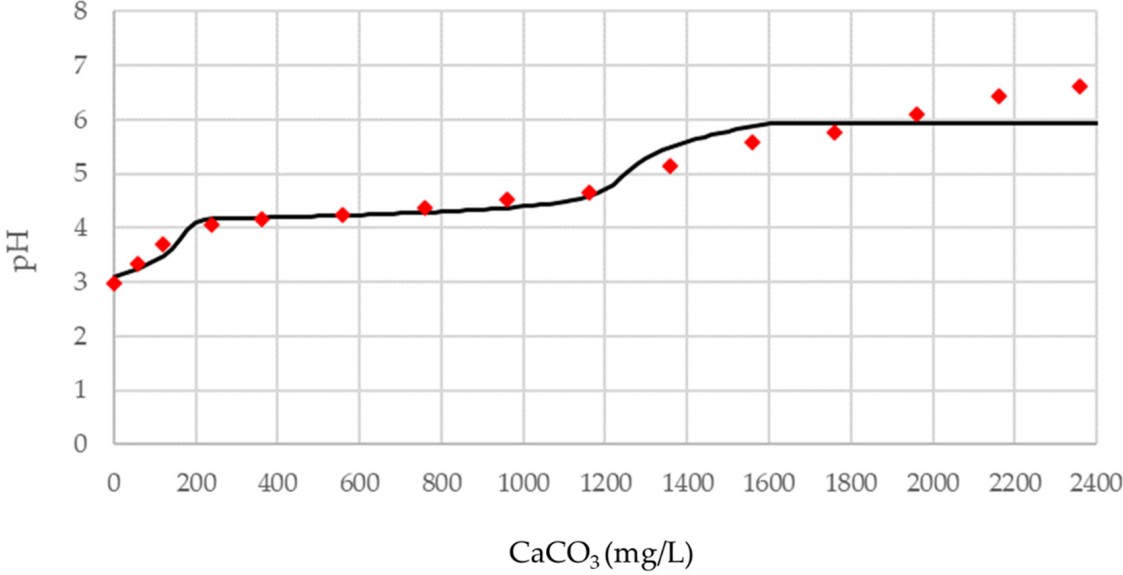

**Figure 11.** Leached buffered sample FA3, modeling geochemical curve (black line) compared with values of buffering test (red diamonds). Liquid samples FA3-S and FA3-L represent the buffered points where respective precipitates were collected. The conceptual scheme just described applies to the graph in Figure 12, referring to sample FA4. The theoretical curve is very different at the same range of buffer concentrations considered, compared with the FA3 solution sample. The acidity of the solution shows stable values over a significantly shorter buffer concentration range, specifically between 120 mg/L and 360 mg/L, corresponding to pH values at which precipitates were formed. The blue circles, which correspond to the amounts of buffering agent added during the test and the respective pH, have a good correspondence up to buffer concentrations of 360 mg/L. Thereafter, there is a shift between the calculated and observed data. Two flat trends can be observed: between 120 mg/L and 360 mg/L and a minor one between 960 mg/L and 1160 mg/L.

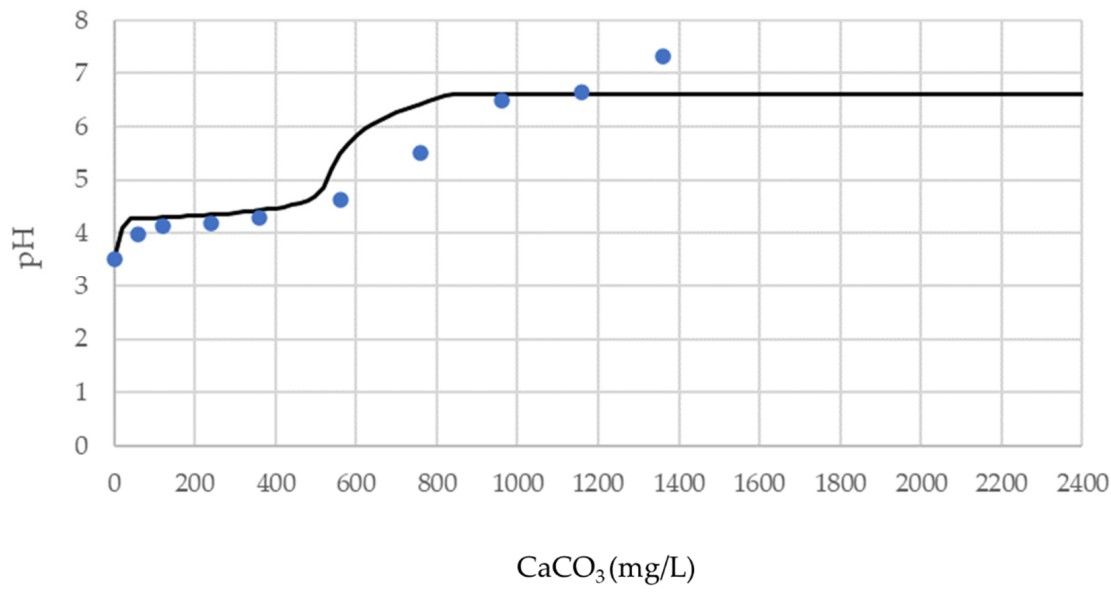

**Figure 12.** Leached buffered sample FA4, modeling geochemical curve (black line) compared with values of buffering test (blue circles). Liquid samples FA4-S and FA4-L represent the buffered points where respective precipitates were collected.

*5.2. Effects of pH Change on Metal Concentrations into Solutions*

The study and characterization of leached and buffered solutions highlights the behavior of metals as pH changes. By measuring the concentration of elements during different buffering steps we can infer which elements are involved in precipitate formation. For this purpose, for each sample, the metal concentrations of the leached and buffered solutions were compared (Figures 13 and 14).

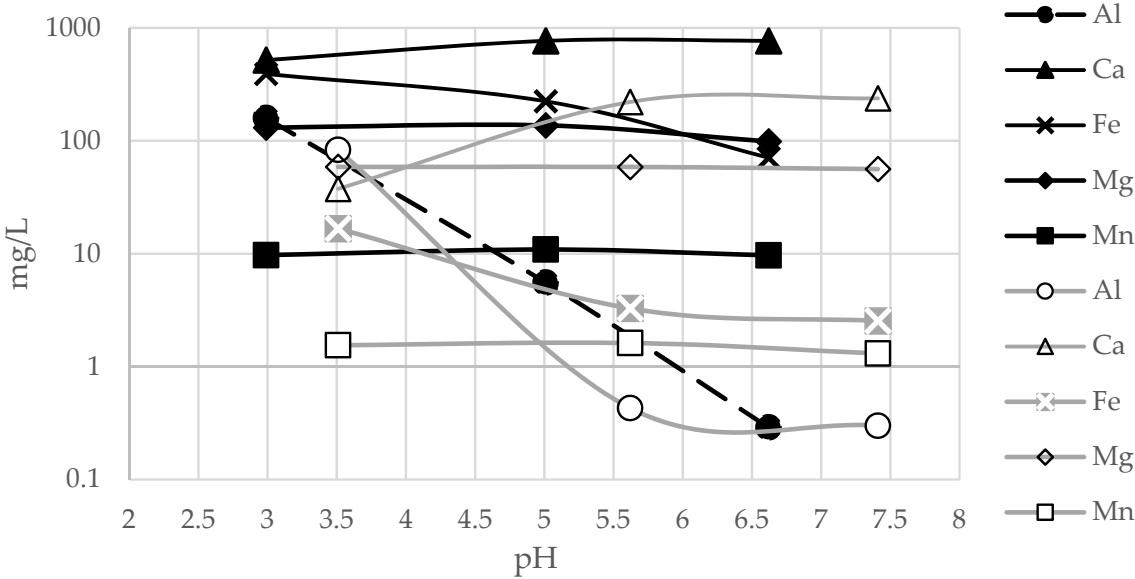

**Figure 13.** Comparison of some major and minor elements ion concentrations between FA3 (full symbols) and FA4 (empty symbols) from the beginning to the end of buffering.

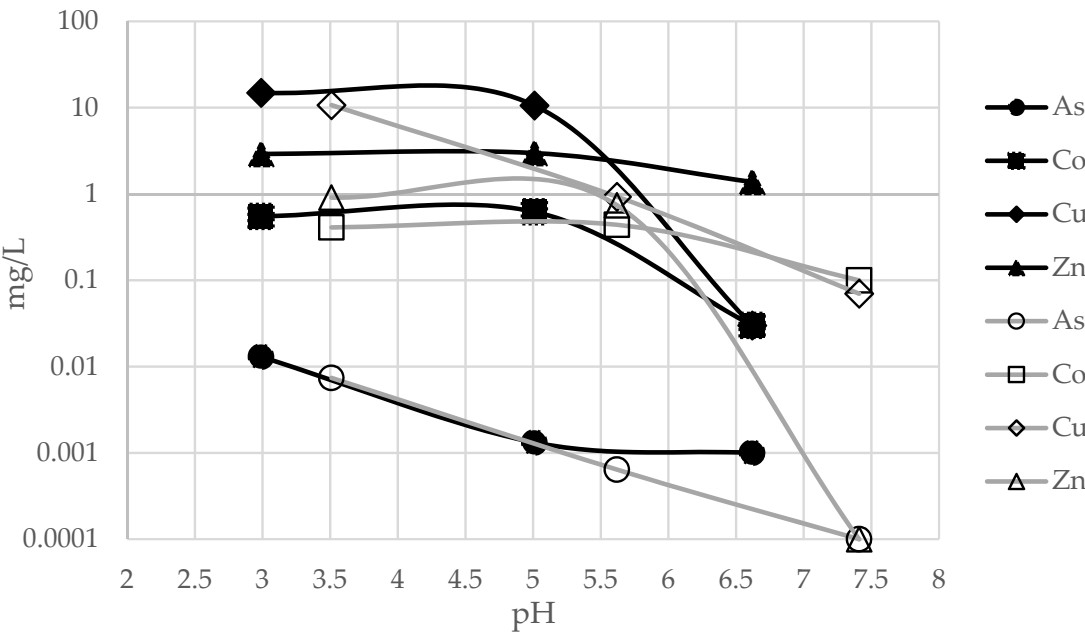

**Figure 14.** Comparison of some trace elements ion concentrations between FA3 (full symbols) and FA4 (empty symbols) from the beginning to the end of buffering.

By comparing Figures 11–14, we can observe a correlation between the changes in the concentration of elements in solution during the buffering steps and specific pH values. In particular, some elements, such as Al, As, Cu and Fe, undergo appreciable concentration decreases along the flat pH stretch at pH = 4–4.5. These results are consistent with the data in the literature [3,12,37–43], showing that there is a correlation between chemical species in solution (and their oxidation state) and their precipitation in specific compounds as a pH function.

For FA3, the decrease in metal concentration between pH 2.99 and 5.01 (see Table 4), is very high for Al (−97%) and As (−90%), and still significant for Fe (−43%) and Cu (−29%), whereas Ca (+48%) increased, being the main component of the buffering agent.

The FA4 sample shows the highest decrease, after short buffering, for Al (−99%), Fe (−80%). Cu (−44%), however, as with FA3, has a high Ca value (+476%) due to the buffering agent.

For both FA3 and FA4 samples, all other ions (Co, Mg, Mn and Zn) have maintained nearly stable values, due to their precipitation at significantly higher pH levels [35–37].

### 5.3. Analysis of Precipitates

The weight of four precipitates, two for each sample, was measured. For FA3 0.020 g (pH = 5.01) and 1.10 g (pH = 6.62) of material have precipitated. Buffered sample FA4 has produced a very modest number of precipitates for both the short and long buffering, at 0.010 g (pH = 5.62) and 0.040 g (pH = 7.41), respectively.

The above points of data show, specifically with regard to sample FA3, that while short buffering only partially reduced the amount of the metals in solution, it produced a minimal amount of sludge. An almost total reduction of metals would produce a formation of sludge 50 times higher.

The precipitated materials, despite their small amounts, were sufficient for a first mineralogical characterization. The results of XRD analyses and TEM analyses equipped by EDS device, showed the presence of mineralogical phases belonging to the families of carbonates, sulfates and various types of hydrated minerals, both hydroxides and silicates, as well as the presence of nano-crystalline to amorphous phases.

XRD analyses of FA3 short buffering precipitates (Figures 3 and 4) show a diversity of mineral phases formed at the respective pH values at which they were collected. The

high initial content in elements such as Al, Ca, Fe and S provided the ingredients for the formation of the mineral phases precipitated and collected at acidic pH (=5.01), dominated by sulfates with subordinate carbonates, hydroxides, and to a lesser extent, silicates and hydrosilicates (Figure 3).

As buffering proceeded to the next step at pH = 6.62, the concentrations of Al, Fe and S decreased, while the concentration of Ca increased. These changes are consistent with the diffraction patterns of the precipitates of the FA3 solution both showing a significant reduction of sulfate phases, whereas carbonate phases stand out and are dominant.

The lower S content in the leachate of FA4 is confirmed by the mineralogy of the precipitates. In the short buffer the sulfate peaks are of modest intensity, and they are completely absent as the buffer proceeds, at pH = 7.41.

The observations are consistent with the results of previous work [3,38], in which both the presented geochemical model and experimental results show precipitation of elements such as Al, Cu, Fe, Mn and Zn in the same pH ranges.

Trace elements such as As, Co, Cu and Zn do not form their own mineralogical phases in precipitates; hence, their behaviour during buffering is harder to track. For this purpose, we opted for a pilot investigation by in-depth TEM analysis, choosing the precipitate with the greatest mineralogical variability, from sample FA3-S. Some significant and representative images of the precipitated material are reported in Figures 7–11.

The results support and confirm the mineralogical classes suggested by the qualitative powder diffraction analyses, enriching them with important information on their chemical composition. The most significant points of data, however, are the size of the crystals, their degree of crystallinity and the confirmation of the presence of amorphous material, as well as its chemical composition. The As, Co, Cu and Zn contents were often found to be below the detection limit of the instrument; thus, only in some cases the presence of elements, e.g., Cu, was detected (Figure 10). The presence of amorphous material also suggests that trace or minor chemical species are not necessarily included in a crystalline structure but may "stick" on the surface of the non-crystalline material. In any case, there is no evidence at the moment that these elements form their own phases.

## 6. Conclusions

The present study confirms that the geochemical modeling of buffering has proven to be very satisfactory in fitting with the experimental tests performed and can be a useful tool in determining buffer settings. These results show the usefulness of the model in defining a priori an optimal amount of buffering agent to be added to the acid solution, minimizing the production of precipitated materials. Calcium carbonate is confirmed to be a suitable buffering agent for both the acidity reduction of the leachate and the removal of metal species in solution: the analysis of leachates from tailing materials shows that the metals in solution have been reduced by 90–100% at pH > 6, with the only exception of Mn.

Leachates from low pyrite tailing materials (FA4) were successfully buffered and metal concentration drastically reduced already with the short test, whereas high pyrite tailing materials (FA3) require a much higher amount of buffering materials, resulting in a higher amount of precipitate.

ICP-MS analyses combined with XRD provide only partial information on the precipitate mineralogy and the behaviour of metals. TEM equipped by EDS proved to be a valuable tool to unravel the processes involved in precipitation and adsorption of metals.

**Author Contributions:** Conceptualization, G.C., G.G. and A.S.; methodology, G.C., A.S. and E.S.F.; software, G.C.; validation, G.C., G.G. and A.C.; formal analysis, G.C., M.B., E.D. and E.S.F.; investigation, G.C., G.G. and A.S.; resources, G.G. and A.C.; data curation, G.C. and M.B.; writing—original draft preparation, G.C.; writing—review and editing, G.C., G.G. and M.B.; visualization, G.C.; supervision, G.G.; project administration, G.G. and A.C.; funding acquisition, G.G. and A.C. All authors have read and agreed to the published version of the manuscript.

**Funding:** This research was funded by the Italian Ministry of Education (MIUR) through the project "PRIN2017—Mineral reactivity, a key to understand large-scale processes and the project 'Dipartimenti di Eccellenza 2017'".

**Acknowledgments:** Authors wish to acknowledge personnel working at Fushe Arrez plant for their support in sampling. We also wish to thank Dott. Nicola Rotiroti and Fernando Camara for their precious technical support about TEM analysis.

**Conflicts of Interest:** The authors declare no conflict of interest.

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
