# Peer review of "Buffering Copper Tailings Acid Mine Drainage: Modeling and Testing at Fushë Arrëz Flotation Plant, Albania"

_water, doi:10.3390/w14152398_

Round 1

Reviewer 1 Report

The manuscript is devoted to the problem of acid drainage buffering of copper-nickel sulfide ores enrichment tailings of the Fushë Arrëz flotation plant, Albania. The addition of commercial CaCO3 paste provided by UNICALCE was investigated. The manuscript is well structured. An important positive point is a good combination of geochemical modeling and experimental tests. But there are a number of remarks:

1)      In order to understand the scale of the problem, a brief description of old and new tailings: its volume, material composition, content of pyrite and other sulfides.

2)      How was FA3 and FA4 sampled? Are these samples averaged?

3)      Is it possible to characterize the mineral phases in more detail (Fig. 4.1-4.4) according to X-ray diffraction data, as well as determine its ratio? Reflexes at a number of angles were not identified. Unknown phases?

4)      Table 3.1 is replaced by Figure 3.1.

Reviewer 2 Report

The proposed article can be interesting for specialists in the area of physicochemical methods of mine drainage treatment and resource recovery. The subject suits the scope of Water. There are some specific comments:

1.       The introduction is recommended to update with information about the other types of treatment of water reach with heavy metals and their leaching from used materials. Following publications can be used for this:

https://doi.org/10.1016/j.jece.2015.04.017

https://doi.org/10.2166/wst.2014.192

2.       P4 L144. 4 rpm is a very low speed. It should be a mistake here.

3.       As a recommendation for future work. This method of mine drainage treatment can be evaluated by life cycle assessment to see if this approach is sustainable and evaluate its environmental impact

4.       The study is very interesting and well methodologically performed. The manuscript is well written but requires some language proof. This work definitely can be accepted after minor revision.
